# Core Endophytic Bacteria and Their Roles in the Coralloid Roots of Cultivated *Cycas revoluta* (Cycadaceae)

**DOI:** 10.3390/microorganisms11092364

**Published:** 2023-09-21

**Authors:** Jiating Liu, Haiyan Xu, Zhaochun Wang, Jian Liu, Xun Gong

**Affiliations:** 1Key Laboratory of Economic Plants and Biotechnology, Kunming Institute of Botany, Chinese Academy of Sciences, Kunming 650201, China; liujiating@mail.kib.ac.cn (J.L.); xuhaiyan@cau.ac.cn (H.X.); wangzhaochun@mail.kib.ac.cn (Z.W.); 2University of Chinese Academy of Sciences, Beijing 100049, China

**Keywords:** coralloid root, endophyte, cyanobacteria, core microbes

## Abstract

As a gymnosperm group, cycads are known for their ancient origin and specialized coralloid root, which can be used as an ideal system to explore the interaction between host and associated microorganisms. Previous studies have revealed that some nitrogen-fixing cyanobacteria contribute greatly to the composition of the endophytic microorganisms in cycad coralloid roots. However, the roles of host and environment in shaping the composition of endophytic bacteria during the recruitment process remain unclear. Here, we determined the diversity, composition, and function prediction of endophytic bacteria from the coralloid roots of a widely cultivated cycad, *Cycas revoluta* Thunb. Using next-generation sequencing techniques, we comprehensively investigated the diversity and community structure of the bacteria in coralloid roots and bulk soils sampled from 11 sites in China, aiming to explore the variations in core endophytic bacteria and to predict their potential functions. We found a higher microbe diversity in bulk soils than in coralloid roots. Meanwhile, there was no significant difference in the diversity and composition of endophytic bacteria across different localities, and the same result was found after removing cyanobacteria. *Desmonostoc* was the most dominant in coralloid roots, followed by *Nostoc*, yet these two cyanobacteria were not shared by all samples. *Rhodococcus*, *Edaphobacter*, *Niastella*, *Nordella*, *SH-PL14*, and *Virgisporangium* were defined as the core microorganisms in coralloid roots. A function prediction analysis revealed that endophytic bacteria majorly participated in the plant uptake of phosphorus and metal ions and in disease resistance. These results indicate that the community composition of the bacteria in coralloid roots is affected by both the host and environment, in which the host is more decisive. Despite the very small proportion of core microbes, their interactions are significant and likely contribute to functions related to host survival. Our study contributes to an understanding of microbial diversity and composition in cycads, and it expands the knowledge on the association between hosts and symbiotic microbes.

## 1. Introduction

The plant root is the interface between multicellular eukaryotes and soil, which is one of the richest microbial ecosystems on Earth [1]. Soil bacteria can colonize roots as benign endophytes and regulate plant growth and development [2], and their influence ranges from improving productivity [3] to phytoremediation [4]. Cyanobacteria are considered to be one of the earliest organisms to dominate the Earth, from the late Archean to early Proterozoic about 3500 to 2700 MYA [5,6,7,8]. These free-living and symbiotic cyanobacteria vary from spherical and cylindrical unicellular forms to filamentous multicellular forms [9], with most filamentous members of *Nostoc* being found to be able to establish symbiotic relationships with hosts for nitrogen fixation [10]. Cyanobacteria have been found to associate with many plants, such as bryophytes, ferns (*Azolla*), gymnosperms (cycads), and angiosperms (*Gunnera*) [11,12]. These symbiotic cyanobacteria found in plants are almost always obtained from the environment (horizontal transfer), except for *Azolla*, in which cyanobacteria are acquired through a vertical transfer process [13].

Cycads, a primitive group of seed plants, include both living and fossil forms that have spanned approximately 300 million years [14,15,16]. Extant cycads are also known as “living fossils” due to their similar morphology to their extinct ancestors. As an ancient relict plant on Earth, cycads show some morphological characteristics that may be linked to adaptation to the adverse environment of drought and poor soil, such as fleshy roots, the formation of periderm, and the evolution of stone cells [17]. The evolutionary conservatism of cycads could also be related to the large number of symbiotic cyanobacteria in their coralloid roots, a type of apogeotropic root that shows repeated dichotomous branching and is nodular [18]. Due to the harsh environment 300 million years ago [19], cycads may have developed a mechanism to resist the barren soil by forming associations with cyanobacteria, with both sides maintaining a mutually beneficial symbiotic relationship [20]. In this context, the host cycads formed a specific coral-like root structure to accommodate cyanobacteria, providing them with a relatively stable living environment and photosynthetic products, and cyanobacteria in return supplied nitrogen nutrition to the host plants through biological nitrogen fixation [21]. The natural nitrogen in the atmosphere does not react with other chemicals, while cyanobacteria can break the triple bond in nitrogen and use nitrogenase to transform inert compounds into useful nitrogen forms to promote plant growth [20,22].

Most previous studies regarding endophytic cyanobacteria in the coralloid roots of cycads were devoted to the discovery of species diversity. Known cyanobacteria that have been identified in cycads include *Nostoc* [23,24,25,26,27,28,29,30], *Anabaena* [26,27], *Calothrix* [23,25,27,29], *Desmonostoc* [30], *Microcoleus*, *Leptolyngbya*, *Chroococcus*, *Scytonema*, *Acaryochloris* [29], *Cylindrospermopsis*, *Trichormus* [27], and *Dolichospermum* [28]. With the development of technology in high-throughput sequencing in recent years, increasing studies on cycad endophytes are shifting to the variation in endophytes in different species, plant tissues, and habitats. For instance, a significant difference was found in the diversity of endophytes between the normal roots and coralloid roots of *Cycas bifida*, but there was no variation when the sequences of cyanobacteria were eliminated, suggesting that cyanobacteria contributed greatly to the differences [28]. For different niches of *Cycas*, the microbial community structure was significantly differentiated, and the biogeography hardly exerted an impact on microbial community variation [31]. Additionally, the study of the microbial diversity of cycad coralloid roots from different sampling sites like cultivated and ex situ plants can also provide insights into the conservation of cycad species with critical endangered status [30]. The above studies have expanded our knowledge that endophyte diversity in host cycads may vary among different species, plant tissues, and niches. Existing evidence shows that there is no species specificity between cycads and cyanobacteria [25]. However, whether the same host species from different environments has a preference for endophytic bacteria in the process of recruitment and the role of recruited core endophytic bacteria remain seldom investigated. 

In this study, we focused on *Cycas revoluta* Thunb. to further explore the role of host *Cycas* in recruiting endophytic bacteria. *C. revoluta* is only native to the islands of southern Japan and Fujian Province of China, but it can adapt to frost and drought habitats in a wide range of areas from tropical to temperate zones. This wide climatic adaptation makes it the most popular and widely cultivated cycad species worldwide for ornament and landscaping [32], thus enabling us to study the variations in bacteria composition across different localities. We used the Illumina MiSeq sequencing approach to reveal the diversity and composition of endophytic bacteria in the coralloid roots and bulk soils of cultivated *C. revoluta* sampled from different locations in China, aiming to answer the following questions: (1) What is the diversity and composition of bacteria in the coralloid roots of *C. revoluta*? (2) What are the core endophytic bacteria and how much variation is there among localities? (3) What are the potential functions of core endophytic bacteria in coralloid roots? 

## 2. Materials and Methods

### 2.1. Sampling

The coralloid roots and associated bulk soils of cultivated *C. revoluta* were gathered from 11 different sites across China during September to November in 2020 (Figure 1). These coralloid roots and bulk soils were retrieved from a depth of less than 10 cm beneath the soil surface and were sourced exclusively from mature and healthy *C. revoluta* plants. In each location, we sampled three coralloid root and bulk soil samples as duplicates. Additional information about the samples can be found in Appendix A. All collected samples were promptly frozen at −20 °C and kept in storage until DNA extraction was performed.

### 2.2. Sample Preparation, DNA Extraction, PCR and Sequencing

Approximately 1 g of mature and healthy coralloid roots was harvested from the plants. Soil debris adhering to the root surface was gently rinsed with running water. Subsequently, the coralloid roots were put through a series of surface sterilization steps. Initially, they were rinsed with 75% ethanol for 1 min, followed by immersion in a 2% commercial bleach solution (NaClO) for 3 min, and then subjected to five subsequent rinses with sterile water. To confirm the effectiveness of surface sterilization, 100 μL of the final rinse water was platted on beef extract peptone medium (BPM) and incubated at 28 °C for 7 days. Samples that were successfully surface sterilized were further processed for DNA extraction using a modified cetyltrimethylammonium bromide (CTAB) method [33]. For bulk soils, 0.5 g of material was prepared for total DNA extraction using the FastDNA^®^ Spin Kit (MP Biomedicals, Santa Ana, CA, USA).

The V4-V5 hypervariable region of the 16S rRNA gene was amplified using the 515F/907R [34] primer pairs. The PCR reaction was carried out in a 20 μL reaction mixture containing 10 ng of template DNA, 0.4 μL of TransStart FastPfu DNA Polymerase, 4 μL of 5X FastPfu Buffer, 2 μL of dNTP (2.5 mM), 0.8 μL of each primer (5 μM), and 0.2 μL of BSA. The PCR amplification consisted of 29 cycles at 95 °C for 30 s, 55 °C for 30 s, and 72 °C for 45 s with a final extension of 72 °C for 10 min. Subsequently, the PCR products were detected and quantified using the QuantiFluor™ -ST blue fluorescence quantitative system (Promega company, Beijing, China), and then mixed according to the sequencing quantity requirements for each sample. The TruSeqTM DNA Sample Prep Kit was employed for library preparation, and the sequencing of the libraries was performed on the Illumina MiSeq PE300 platform at Shanghai Majorbio Bio-pharm Technology Co., Ltd. (Shanghai, China).

### 2.3. Sequence Processing and Statistical Analyses

The Illumina MiSeq sequencing generated raw data, which were then associated with each sample. Subsequently, the paired-end raw reads underwent quality filtering using fastp v0.19.6 and were further assembled using FLASH v1.2.7, based on the overlapping regions between paired-end reads. The resulting high-quality data were subjected to denoising through the DADA2 [35] process within the QIIME2 software (version 2020.2), resulting in the acquisition of Amplicon Sequence Variant (ASV) representative sequences and their corresponding abundance information. Taxonomic information for each representative sequence was assigned using the silva138/16s_bacteria database (https://www.arb-silva.de/, accessed on 19 April 2021).

Prior to conducting diversity analysis, ASVs identified as chloroplasts and mitochondria were filtered out. Mothur-1.30 [36] were used to calculate alpha diversity index and visualize rarefaction curves. Specifically, we computed Chao 1 [37], Shannon [38], and Simpsoneven indices [39] to access the complexity of species diversity within each sample. The differences in these indices among samples were performed based on a Kruskal–Wallis test in R v4.0.4. To explore potential variations in bacterial community composition among sampling sites, principal co-ordinates analysis (PCoA) was carried out using the R package “vegan (v2.5-7)” [40]. This analysis was based on the Unweighted UniFrac distances at the ASV level between coralloid roots and bulk soils. Furthermore, bar plots were generated to visualize the relative abundance of microbial communities among samples, utilizing the R package “tidyverse” [41] and “ggplot2” [42]. 

We utilized PICRUSt2 (v2.2.0-b) to predict the function of bacteria community in different samples [43]. To detect variations in bacterial function between coralloid roots and bulk soils, we employed Welch’s t-test and generated extended error bar plots using STAMP (v2.1.3) [44]. The identification of stable and permanent endophytes within the coralloid roots of *C. revoluta* was accomplished through two distinct approaches. Firstly, we considered shared taxa across all samples as the core microbiome, represented by the overlapping portions in a Venn diagram. This analysis was conducted using flower plots via the website (http://www.ehbio.com/test/venn/#/, accessed on 12 May 2023) [45]. Secondly, we defined core endophyte taxa based on network topological characteristics within a microbial co-occurrence network. To achieve this, we calculated Spearman’s correlation coefficient and their significance using the “psych” package in R (v4.0.4) [46]. The co-occurrence patterns of the bacterial community within coralloid roots were explored through network inference using the Gephi software (version 0.10.1). Strong and significant correlations (R > 0.6, *p* < 0.05) were considered for this analysis [47].

## 3. Results

### 3.1. Sequence Metrices and Diversity Analysis of Bacteria in Coralloid Roots and Bulk Soils

The bacterial communities associated with both coralloid roots and bulk soils of *C. revoluta*, sampled from 11 different sites, were subjected to analysis. In total, 1,312,987 high-quality reads were generated, with 532,247 originating from endophytes and 780,740 from bulk soils. Clustering these sequence reads resulted in a total of 16,568 ASVs. The soil samples exhibited significantly higher microbial diversity, comprising 16,388 ASVs, in contrast to the endophytes, which showed a more limited diversity with only 354 ASVs detected. Rarefaction curves were constructed for each individual sample, based on observed ASVs (Appendix A), revealing that ASV richness per sample reached saturation with increasing sequencing depth, affirming the adequacy of the data volume of sequenced reads.

Upon aligning the obtained ASVs with various levels of bacterial taxa (Appendix A), we observed that a substantial 98.86% of the total sequence reads could be categorized into 42 bacterial phyla. Of these, 16 phyla were identified in coralloid roots, while all 42 phyla were present in the soil samples. At the genus level, we successfully assigned 81.64% of the bacterial sequences, revealing a total of 1137 genera, with 177 found in endophytes and 1121 in bulk soils. When delving into lower taxonomic rankings, only a minimal 0.08% and a more notable 20.61% of the total reads could be further classified to the species level in coralloid roots and bulk soils, respectively. 

Alpha diversity in the bulk soils greatly exceeded that in the coralloid roots for bacteria communities across all 11 sampling sites (Table 1). Notably, when examining community richness, evenness, or diversity, which were represented by chao1 index, Shannon index, and Simpsoneven index, respectively, no significant differences were observed among both coralloid roots and bulk soils (*p* > 0.05, Table 1). This lack of significance persisted even when cyanobacteria were excluded from the analysis of coralloid roots (*p* > 0.05) across all 11 sites. 

The PCoA results of Adonis-based inter-group difference test revealed that there was no statistically significant difference in the bacterial community composition of coralloid roots among the 11 sites (Figure 2a). This result remained unchanged even after the exclusion of cyanobacteria from the analysis (Figure 2b). However, a significant difference was observed when comparing the bacterial community composition between the bulk soil and coralloid root samples (Figure 2c), as well as among the different bulk soil samples (Figure 2d). This suggested that the composition of the bacterial community within the soils was significantly influenced by geographical or environmental factors.

### 3.2. Variation in Bacterial Composition

A total of 98.86% of the sequences generated in this study were successfully classified at the phylum level. Among the identified phyla, *Cyanobacteria* stood out as the dominant group in coralloid roots, with a remarkable relative abundance of 99.27%. However, they exhibited exceptionally low diversity, accounting for only 4.71% of the total ASVs. In contrast, *Proteobacteria* and *Actinobacteriota* each comprised less than 1% of the community (Appendix A), but these two phyla collectively contributed significantly to the overall diversity, accounting for 54.92% and 12.06%, respectively. In the soil bacterial assemblage, *Actinobacteriota* was the most abundant, comprising 31.40% of the community, followed by *Proteobacteria* (23.11%), *Firmicutes* (13.24%) and *Acidobacteriota* (11.46%). Interestingly, *Cyanobacteria* exhibited a notably low abundance of 0.31% in the soil, suggesting that they were selectively recruited by the host and enriched in the coralloid roots of *C. revoluta* (Figure 3).

At the family level, approximately 96.87% of reads could be identified and assigned to 531 families across the samples, with 107 families observed in coralloid roots and 524 in bulk soils. *Nostocaceae* from *Cyanobacteria* was the most dominant family of endophytic bacteria in coralloid roots, with an average abundance of 99.03%. In contrast, the abundances of endophytic bacteria in all other families were notably lower, typically below 0.20%, including families such as *Burkholderiaceae* (0.16%) and *Nocardiaceae* (0.16%). In bulk soils, the most abundant family was *Bacillaceae*, accounting for 6.16% of the community, followed by *Xanthobacteraceae* (3.57%), *Burkholderiaceae* (2.82%), *Micrococcaceae* (2.78%), *Nocardioidaceae* (2.57%), *Gaiellaceae* (2.47%) and *Solirubrobacteraceae* (2.43%).

At the genus level, approximately 28.80% of the sequences classified as *Nostocaceae* could not be further resolved to specific genera. Notably, *Desmonostoc*, observed in nine sampling sites, was highly abundant in the coralloid roots, constituting 61.68% of the total abundance. *Nostoc* was exclusively found in FZ, XM and YY, with an average abundance of 8.26%. The remaining identified genera of endophytic bacteria in coralloid roots exhibited relatively low abundance, typically less than 0.05%, including genera such as *Tolypothrix* (*Nostocaceae*, 0.27%) and *Rhodococcus* (*Nocardiaceae*, 0.16%). In bulk soils, the most abundant genus was *Bacillus*, representing 6.06% of the community, followed by *Gaiella* (2.47%), *Burkholderia*-*Caballeronia*-*Paraburkholderia* (2.29%), *Nocardioides* (1.84%), *Paenibacillus* (1.27%), *Streptomyces* (1.10%), *Acidothermus* (1.10%) and *Solirubrobacter* (1.06%). It’s worth noting that the majority of these abundant genera in soils belong to the *Actinobacteriota*, with the exceptions being *Burkholderia*–*Caballeronia*–*Paraburkholderia* and *Paenibacillus*.

### 3.3. Function Variations between Coralloid Roots and Bulk Soils

The COG function classification for all samples in this study revealed 22 potential functional categories (Appendix A). Notably, in the COG functions of endophytic bacteria within coralloid roots, the highest proportions were attributed to functions related to amino acid transport and metabolism, inorganic ion transport and metabolism, and energy production and conversion. In bulk soils, the dominant functions were associated with amino acid transport and metabolism, energy production and conversion and translation, ribosomal structure, and biogenesis. 

The Wilcoxon rank-sum test revealed significant differences in all 22 COG functions between coralloid roots and bulk soils as depicted in Figure 4. Regardless of whether it was coralloid roots or bulk soils, functions related to amino acid transport and metabolism consistently ranked highest among the 22 functions. A similar pattern was also found in functions related to glyoxalase bleomycin resistance protein dioxygenase (Appendix A). However, the functions ranking second and third in terms of proportion differed between coralloid roots and bulk soils (Appendix A). In coralloid roots, the most predominant COG functions were related to inorganic ion transport and metabolism, which could be further specified as the phosphate abc transporter and the rieske 2Fe-2S domain-containing protein. Conversely, in bulk soils, the top three COG functions in the category of energy production and conversion were associated with aldo-keto reductase, ferredoxin and catalyzing the hydroxylation to form hypusine, as detailed in Appendix A.

### 3.4. Core Microorganisms in Coralloid Roots of C. revoluta

Based on the membership analysis results, the number of genus unique and shared among different sites were calculated and visualized (Figure 5a). A total of 156 genera were identified, with just one genus, *Rhodococcus* (*Nocardiaceae*, belonging to *Actinobacteriota*), being shared across all 11 sites. Remarkably, *Rhodococcus* accounted for a mere 0.17% of the total relative abundance. However, it is worth noting that these findings appeared inconsistent with the results of the network analysis. The microbial network in coralloid roots consisted of 144 nodes (genus) and 1354 edges (Figure 5b). According to the network connectivity statistics (Appendix A), six genera, i.e., *Edaphobacter*, *Niastella*, *Nordella*, *SH*-*PL14*, *Virgisporangium*, and an unclassified taxon from *Vicinamibacteria* (*Acidobacteriota*), were defined as the core microbiome. Despite representing only 0.01% of the total sample abundance, these core microbes exhibited stronger interactions with each other compared to the remaining genera (Appendix A).

## 4. Discussion

### 4.1. Associations between the Hosts and Endophytic Bacteria

The diversity and composition of root endophytic bacteria communities are primarily influenced by factors such as host genotype, geographical location, soil source, and cultivation practices [48]. In the course of our study, we observed that the alpha and beta diversity of bacterial communities in coralloid roots remained relatively consistent across different geographical locations, regardless of the presence or absence of cyanobacteria in these roots. This finding strongly indicates that the species and quantities of bacteria within coralloid roots are more likely to be shaped by the characteristics of the host plant, *C. revoluta*, rather than environmental variables. This aligns with the outcomes of numerous prior investigations involving cycads and other plant species, all of which suggest that the composition of root endophytic bacterial communities can be substantially influenced by the genetic makeup of the host plant [27,28,29,30,31,49,50].

The above finding implied that host plants may exhibit a preference in selecting symbiotic endophytic bacteria during their recruitment process. Conversely, our findings also indicate that these endophytic bacteria may contribute to the growth and development of the host plant. Their functions appear to be closely linked to amino acid metabolism, inorganic ion transport and metabolism, as well as energy production and conversion within the hosts. More specifically, the endophytic bacteria are associated with functions such as the phosphate abc transporter and rieske 2Fe-2S domain-containing protein within the category of inorganic ion transport and metabolism. This suggests that these endophytes may have a role in the uptake of phosphorus and metal ions, potentially contributing to plant disease resistance [51]. Furthermore, previous research has shown that *Actinobacteria* tend to become enriched in plant roots [52], rhizosphere [53], and soils [54] under drought stress conditions. Although the exact function of *Actinobacteria* detected in both coralloid roots and bulk soils of *C. revoluta* remains uncertain, we hypothesize that it may be linked to enhancing the host plant’s resistance to drought stress.

The process of assembling microorganisms in plant roots is intricate, with two distinct pathways regarding the origin and transmission of bacteria from roots [55,56]. The first pathway involves vertical transfer via seeds, wherein endophytic bacteria initially present in the seeds colonize the endosphere of the developing host plant. These bacteria subsequently reach the plant’s reproductive organs during seed formation and repeat the whole cycle. The second pathway is horizontal transmission, wherein the majority of bacterial endophytes are horizontally transmitted from soil-borne microorganisms. They gain entry to the interior of the plant root through openings such as cracks in root hairs or emerging lateral roots [57]. An illustration of the horizontal transmission pathway can be observed in the case of *C. panzhihuaensis*, where researchers discovered a higher level of species diversity in seeds compared to roots. However, there was limited overlap in bacterial taxa between the two [31], suggesting that the majority of bacteria in the roots likely originate from the surrounding soil.

In our study, we observed a markedly higher diversity of microorganisms in the soil compared to that in the coralloid roots, and there is a significant difference between them. This finding lends support to the hypothesis of a horizontal transmission pathway, given the substantial discrepancy in bacterial diversity between the two compartments. It is worth noting that our experimental samples were primarily collected from ex situ gardens, where frequent human activities and interventions occur. Interestingly, despite the presence of these human disturbances, we did not observe significant variations in the alpha diversity of bulk soils among the sampling sites (Figure 1d). However, there was a notable difference in beta diversity. One plausible explanation for this pattern is that, despite the varying environmental factors such as geography and climate across different locations, the soil niches in these areas remained relatively stable due to human interventions, such as fertilization and watering practices. Within this context, geo-graphic factors may have contributed to the differences in bacterial composition observed in bulk soils, and this in turn might explain the presence of similar, though not identical, bacterial communities in coralloid roots sampled from different locations. It is possible that these variations in both bulk soils and coralloid roots collectively contribute to enhancing the host plant’s adaptability to the local environment [58]. Furthermore, these findings suggest that the composition of the bacterial community in coralloid roots is influenced by both the host plant and the surrounding environment, with the host playing a more prominent role in shaping this composition.

### 4.2. Cyanobacteria Are Dominant in Coralloid Root

In the realm of classified phyla, *Cyanobacteria* emerge as the dominant inhabitants of coralloid roots, boasting a staggering relative abundance exceeding 99%. However, their prevalence in these roots contributes only a modest portion, less than 5%, to the overall species diversity. This observation aligns consistently with prior research findings [24,25,27,59,60,61]. Several hypotheses have been advanced to elucidate the reasons behind the overwhelming dominance of *Cyanobacteria* within coralloid roots [28]. First and foremost is the concept of historical superiority associated with cyanobacteria. The presence of a species can significantly influence the subsequent composition of a community. *Cyanobacteria*, as one of the most ancient life forms, can be traced back approximately 16.5 to 14 million years ago, as supported by fossil evidence [62]. Furthermore, this group exhibits robust environmental adaptability, low host specificity, and rapid mobility, allowing it to swiftly infiltrate hosts and establish itself within an optimal internal habitat. A second explanation lies in the unique structure of cycad coralloid roots, which facilitates cyanobacterial invasion. Unlike regular roots, which possess a thick periderm and structured vascular tissue that serve as barriers to cyanobacterial intrusion [17], coralloid roots feature a thin periderm, an outer cortex, and a thick inner cortex [63]. This structural distinction not only eases cyanobacterial infiltration but also limits their overabundance. Thirdly, cyanobacteria serve as a vital nitrogen source for cycads, fostering mutualistic symbiosis between the two. Lastly, the fourth hypothesis centers on the secondary metabolites generated by the symbiotic relationship between cycads and cyanobacteria. These metabolites, including phenolic compounds [64], polysaccharides [65], and cyanobacterial toxins [25,66,67], exert control over the invasion of other microorganisms, solidifying cyanobacteria’s dominant position within coralloid roots [25].

Furthermore, our findings unveiled that the *Nostocaceae* comprised the predominant group of endophytic bacteria in coralloid roots, with an average abundance of 99.03%. At the genus level, *Desmonostoc* exhibited a high prevalence at 61.68%, followed by *Nostoc* (8.26%), making them the most prevalent cyanobacteria in wild *C. revoluta* coralloid roots [27]. Although *Desmonostoc* was detected in cycad coralloid roots, it did not hold the same dominant status as a cyanobacterial species [30].

### 4.3. Core Bacteria in Coralloid Roots of C. revoluta and Potential Ecological Functions

The core microbiome, which represents a distinctive cluster of microorganisms, is an emerging and promising research area offering fresh avenues for enhancing the growth and productivity of a host organism [68]. In our study, we employed two methods, namely membership and network connection analysis, to investigate the essential endophytic communities within the coralloid roots of *C. revoluta*. 

*Rhodococcus* was the only genus found to be consistently present across all sampling sites. This genus has also been recognized as a core endophyte in *Jasione montana* [69]. Notably, various species within the *Rhodococcus* genus, including *R. aetherivorans*, *R. erythropolis*, *R. equi,* and *R. rhodochrous*, are known for their beneficial roles in enhancing plant growth and exhibiting resistance to arsenic [70,71,72].

According to the network analysis, the core genera identified were *Edaphobacter*, *Niastella*, *Nordella*, *SH-PL14* and *Virgisporangium*. *Edaphobacter*, which is affiliated with *Acidobacteriaceae* (*Acidobacteriota*), has been primarily linked to alpine and forest soils [73,74]. Additionally, *Edaphobacter* has shown associations with smut resistance in sugarcane [75]. In the context of plant endophytes, *Edaphobacter* was found to be more abundant in the roots than in the bulk soil and rhizosphere of cultivated *Allium ulleungense* [76]. *Niastella* is categorized within the *Bacteroidota* class, specifically in the *Chitinophagaceae.* All currently known species from this genus have been isolated from soil and can be cultured on R2A agar plates [77,78,79]. In soil environments, the majority of species within this genus exhibit multiple antibiotic resistance, tolerance to various metals, and the capability to hydrolyze chitosan [80]. Notably, *Niastella* has been found to be highly abundant in the lateral roots of sugar beet phytosphere [81], and it has been reported to have a higher presence in sugarcane roots compared to bulk soils [82]. This suggests that *Niastella* may be particularly attracted to the rhizosphere of high sugar-accumulating crops [81]. *Nordella*, a member of the *Rhizobiales*, has been identified within the interior of healthy cowpea root nodules (*Vigna unguiculata* L. Walp.) [83] and in the stems of transgenic poplar trees [84]. *Nordella* exhibits the capacity to colonize the rhizosphere of maize and soybean [85], and it is often associated with soil water content, nitrogen fixation, and the decomposition of organic matter. This bacterial genus has been recognized as a significant contributor to nutrient cycling in forest ecosystems [86], highlighting its potential as a valuable soil microorganism [87,88]. *SH-PL14*, belonging to *Planctomycetota* and *Rubinisphaeraceae*, is distinguished by its ability to perform anaerobic oxidation of NH_4_^+^ to N_2_, a process known as anammox [89,90]. *Virgisporangium* belongs to the *Actinobacteriota*, specifically within the *Micromonosporaceae*, and it is frequently found in tropical and subtropical soils [91]. Moreover, *Virgisporangium* has been identified as an endophyte in the roots of various plants, including desert shrubs [92], wheat [93], grass [52], and *Asparagus officinalis* [94]. Notably, the abundance of *Virgisporangium* has been observed to significantly increase when the host plants were treated with gibberellic acid (GA) [95] and nitrogen [96], respectively.

As a result, the identification of these six bacterial genera, *Rhodococcus*, *Edaphobacter*, *Niastella*, *Nordella*, *SH-PL14* and *Virgisporangium*, as core microorganisms within cycad coralloid roots represents a novel discovery. These findings shed new light on the crucial microbial inhabitants of cycad root systems. Given the potential interactions and roles of these core microbes in coralloid roots, future research should prioritize investigating their contributions to the growth and resilience of host cycads, as well as their involvement in coping with adverse environmental conditions.

## 5. Conclusions and Future Perspectives

In this study, we demonstrated the advantages of employing NGS sequencing to investigate species diversity and community composition of bacteria in both coralloid roots and bulk soils of *C. revoluta*. The significantly greater diversity of bacteria observed in bulk soils, albeit distinctly different from that in coralloid roots, underscores the host’s preference for recruiting microorganisms to the coralloid roots through horizontal transfer. During this process, the host tends to selectively enlist bacteria that provide beneficial advantages for their survival. Moreover, the relatively uniform community diversity and composition of endophytic bacteria across different geographical locations suggest that both the host plant and its surrounding environment contribute to the recruitment of endophytic bacteria. In this context, the host appears to exert a more pronounced influence than the environmental factors. Future research endeavors should consider expanding the sample and adopting comprehensive metagenomic tools to delve deeper into the intricate interactions between cycads and their endophytes. Furthermore, it is essential to explore the ecological roles of core endophytes.

Additionally, given the nitrogen fixation abilities of cyanobacteria and the ecological functions of other core microorganisms, their potential applications in agriculture, such as biofertilizers, antimicrobial agents, and other valuable products, should be explored. The research and development of transgenic crops that can coexist harmoniously with cyanobacteria hold promise for enhancing the sustainability of agriculture.

## Figures and Tables

**Figure 1 microorganisms-11-02364-f001:**
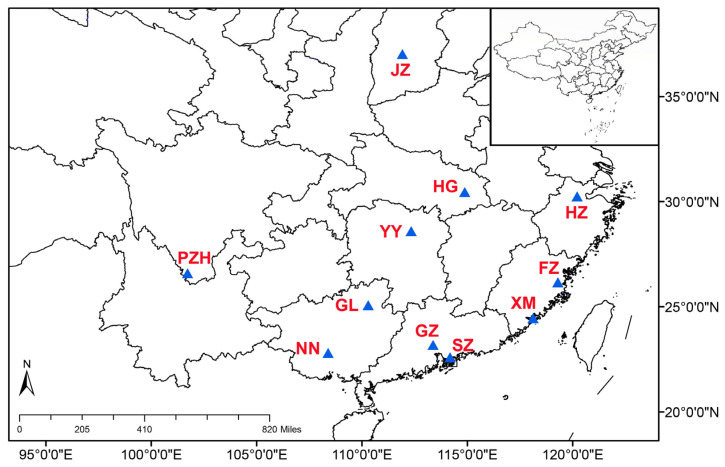
The geographic distribution of sampling sites (blue triangle) in China; the capital letter represents the abbreviation of sampling localities.

**Figure 2 microorganisms-11-02364-f002:**
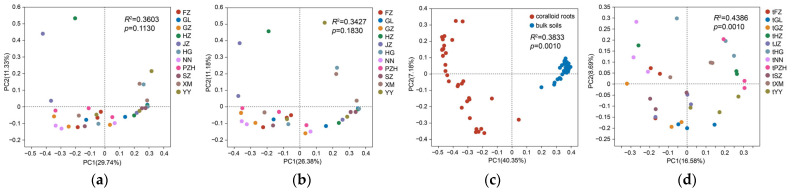
Principal co-ordinates analyses results of bacteria community composition among different sample types of *Cycas revoluta* at ASV level. (**a**) Community in coralloid roots; (**b**) community in coralloid roots after removing cyanobacteria; (**c**) community between coralloid roots and bulk soils; (**d**) community in bulk soils.

**Figure 3 microorganisms-11-02364-f003:**
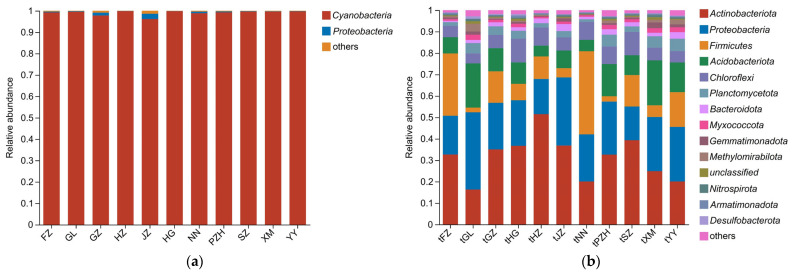
Taxonomic composition of bacteria in coralloid roots (**a**) and bulk soils (**b**) of *Cycas revoluta* from different sampling localities. Phyla with relative abundance less than 0.01% are merged into “others”.

**Figure 4 microorganisms-11-02364-f004:**
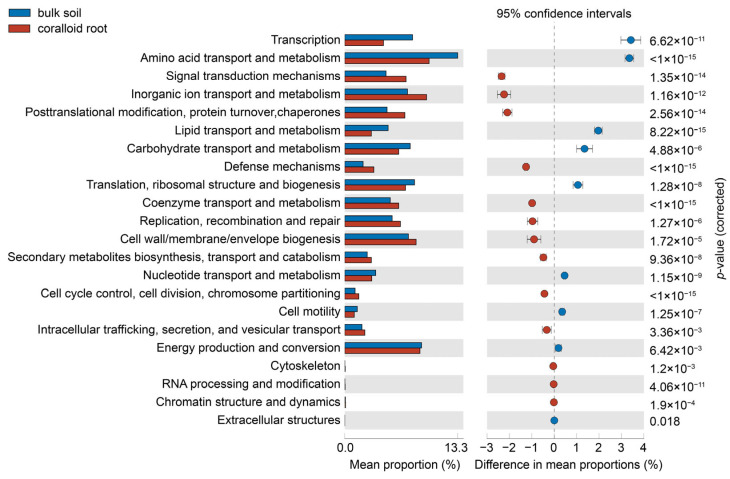
Extended error bar plot of bacterial communities in coralloid roots and bulk soils. The COG functions are listed to the left.

**Figure 5 microorganisms-11-02364-f005:**
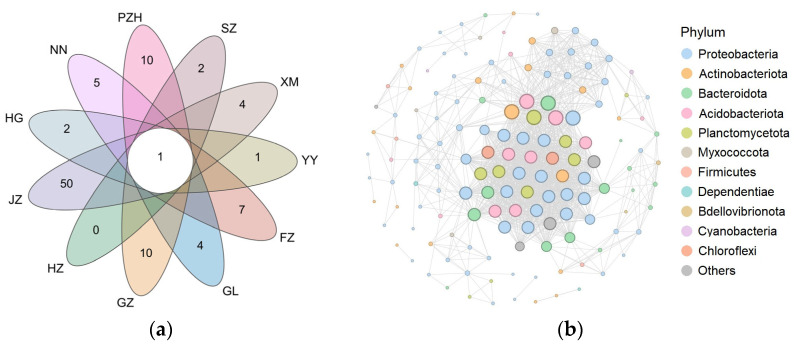
Core microbes in coralloid roots of *C. revoluta* at the genus level. (**a**) Flower plot demonstrated the endophytic bacteria shared among 11 sites; (**b**) correlation co-occurrence network of endophytic bacteria from *C. revoluta* coralloid root. Nodes represent genera; node size represents degree; node color represents the belonged phylum of genus; edges represent Spearman’s correlation (*R* > 0.6, *p* < 0.05).

**Table 1 microorganisms-11-02364-t001:** The alpha diversity indices of different samples in this study.

The Niche Bacteria Exist	Alpha Diversity Index	FZ	GL	GZ	HG	HZ	JZ	NN	PZH	SZ	XM	YY	*p* Value
coralloid roots	Chao	15.88	17.50	21.92	6.92	4.83	57.42	23.70	22.55	8.00	11.56	9.08	0.14
	Shannon	0.30	0.28	0.21	0.06	0.06	0.33	0.14	0.43	0.07	0.43	0.07	0.29
	Simpsoneven	0.13	0.13	0.07	0.03	0.03	0.08	0.04	0.15	0.04	0.20	0.05	0.43
coralloid roots (without cyanobacteria)	Chao	13.64	15.83	20.58	5.25	3.50	55.75	21.56	20.04	6.67	9.56	7.75	0.15
	Shannon	2.08	1.29	2.46	0.95	1.02	2.71	2.36	2.34	1.39	1.51	1.06	0.17
	Simpsoneven	0.54	0.51	0.55	0.43	1.00	0.45	0.58	0.61	0.76	0.82	0.70	0.21
Bulk soil	Chao	747.70	832.30	868.80	1075.00	835.50	882.00	929.30	1112.00	985.60	1058.00	1072.00	0.41
	Shannon	5.47	6.11	6.09	6.34	6.02	5.85	5.40	6.55	6.32	6.56	5.99	0.15
	Simpsoneven	0.17	0.31	0.19	0.29	0.20	0.17	0.11	0.43	0.31	0.47	0.33	0.10

## Data Availability

The raw data of sequencing are available in the NCBI Sequencing Read Archive (SRA) with the accession BioProject ID: PRJNA1007426.

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
