# Peer review of "Core Endophytic Bacteria and Their Roles in the Coralloid Roots of Cultivated Cycas revoluta (Cycadaceae)"

_microorganisms, 2023, doi:10.3390/microorganisms11092364_

Round 1
Reviewer 1 Report
The article that has been submitted for review is undeniably captivating and holds the potential to make a substantial contribution to the crucial discourse surrounding the symbiotic relationships between higher plants and microorganisms. By delving into this intricate interplay, the article sheds light on fundamental ecological dynamics that shape our natural world. The authors have harnessed cutting-edge progress in analytical methodologies and data visualization, elevating the precision and clarity of their findings.
While the article demonstrates an impressive command of scientific methodology and theoretical underpinnings, it is worth noting that the exploration of practical implications stemming from the research remains somewhat limited. The application of scientific discoveries in tangible, real-world scenarios is a bridge that holds immense promise. By extending the analysis to discuss how these newfound insights might translate into actionable strategies, industries such as agriculture, biotechnology, and sustainable resource management could stand to gain. This would not only accentuate the scholarly significance of the work but also reinforce its relevance in addressing contemporary challenges.
The integration of practical considerations doesn't merely reside in the realm of applied science—it underscores the importance of research in our everyday lives. The inclusion of potential applications and their associated benefits, as well as potential challenges or ethical considerations, not only enriches the article's scope but also invites a wider audience to appreciate the broader impact of scientific inquiry. By embracing the dual nature of research—to inform both theoretical constructs and tangible applications—the authors have an opportunity to forge a more holistic narrative that resonates across academia, industry, and society at large.
A thorough linguistic and stylistic revision should elevate your article to an even higher level.
Only small stylistic corrections are required (English).
Reviewer 2 Report
Authors of the article "Core endophytic bacteria and their roles in coralloid roots of cultured Cycas revoluta (Cycadaceae)" used the popular and affordable NGS sequencing approach. Unfortunately, the lack of other data in manuscript, for example, the physicochemical parameters of the environment does not allow to make any conclusions about the effect of something on the microbiome.
The main comment: The authors described in detail the procedure of sterilizing of surface of the roots. As evidence of sterility culturing on the medium was carried out. The absence of bacterial growth is taken as evidence of sterility. But it is known that only a small number of bacteria can be cultured. So, such proofing of sterilization is untruthworthy. In addition, sterilizing agents can penetrate into root and kill the microorganisms.
I have some questions and proposals.
1. The authors did not specify why these 11 samples were taken? are they have the same properties or different? For reader it would be more comfortable to see the map, since it is not clear from the table how far the points are from each other.
2. How many g of soil and root were taken for DNA isolation? from what depth? This information is absent in manuscript.
3. In Figure 2, the authors should make the Actinobacteriota a different color than Сyanobacteria for better visualization.
Line 99: How is the diversity and composition of bacteria in coralloid roots of C. revoluta; - The sentence is unclear.
